# Low melt viscosity enables melt doublets above the 410-km discontinuity

Longjian Xie [1,2] ✉, Denis Andrault [3], Takashi Yoshino [4], Cunrui Han[5], James O. S. Hammond[5], Fang Xu[6], Bin Zhao[4], Oliver T. Lord [7], Yingwei Fei [8], Simon Falvard[3], Sho Kakizawa [9], Noriyoshi Tsujino[9], Yuji Higo[9], Laura Henry [10], Nicolas Guignot[10] & David P. Dobson [2] ✉

Seismic and magnetotelluric studies suggest hydrous silicate melts atop the 410 km discontinuity form 30–100 km thick layers. Importantly, in some regions, two layers are observed. These stagnant layers are related to their comparable density to the surrounding mantle, but their formation mechanisms and detailed structures remain unclear. Here we report a large decrease of silicate melt viscosity at ~14 GPa, from 96(5) to 11.7(6) mPa·s, as water content increases from 15.5 to 31.8 mol% $H_2O$. Such low viscosities facilitate rapid segregation of melt, which would typically prevent thick layer accumulation. Our 1D finite element simulations show that continuous dehydration melting of upwelling mantle material produces a primary melt layer above 410 km and a secondary layer at the depth of equal mantle-melt densities. These layers can merge into a single thick layer under low density contrasts or high upwelling rates, explaining both melt doublets and thick single layers.

The seismic discontinuity at 410-km depth, attributed to the pressure-induced transformation of the main mantle mineral olivine to wadsleyite, marks the bottom of the upper mantle. Just above the 410-km discontinuity, both seismic and magnetotelluric studies have revealed the presence of anomalies at varying depths[1–7]. The primary seismic method for characterizing these anomalies is the receiver function, which is particularly sensitive to rapid changes in seismic velocities through converted seismic phases[2,3,6,7]. Global studies identified seismic low-velocity layers (LVLs) above 410 at ~60% of stations across diverse geodynamic environments, indicating a global distribution (Supplementary Fig. 1a)[2]. The absence of LVLs at the remaining 40% of stations may be due to the limitations of global analyzes, which are unable to resolve layers thinner than 20 km because of high noise levels[2]. LVLs exhibit significant thickness variations, ranging from 30 to 100 km, with lateral changes of over 20 km observed between stations less than 200 km apart. Additionally, although not explicitly

highlighted in previous work, some stations show signals of LVL doublets, consisting of two thin LVL layers separated by a few tens of kilometers, located between 300-km depth and the 410 (Supplementary Fig. 1b)[2]. The existence of LVL doublet is further supported by regional studies[3,6], such as those in Afar[6] (Supplementary Fig. 2), where high-quality, relatively high-frequency waveforms (~0.5 Hz) enable resolving layers as thin as 10 km.

These LVLs are frequently explained by the presence of up to a few wt% of hydrous silicate melt (hereafter referred to as "410 melt")[8–12]. Understanding migration and storage of melt in polycrystalline rocks is therefore key to deciphering the formation mechanisms of these layers. The melt composition generated by dehydration melting of hydrated mantle transition zone is ultramafic[13,14]. These melts exhibit extremely low dihedral angles with olivine above 410, indicating that even a small amount of melt (less than 0.1 vol%) is sufficient for melt migration by percolation and

[1]Center for High Pressure Science & Technology Advanced Research, Shanghai, China. [2]Department of Earth Sciences, University College London, London, UK. [3]Université Clermont Auvergne, CNRS, IRD, OPGC, Laboratoire Magmas et Volcans, Clermont-Ferrand, France. [4]Institute for Planetary Materials, Okayama University, Misasa, Tottori, Japan. [5]School of Natural Sciences, Birkbeck, University of London, London, UK. [6]School of Earth Sciences, Zhejiang University, Hangzhou, China. [7]School of Earth Sciences, University of Bristol, Bristol, UK. [8]Earth & Planets Laboratory, Carnegie Institution for Science, Washington DC, USA. [9]Japan Synchrotron Radiation Research Institute, Sayo, Hyogo, Japan. [10]Synchrotron SOLEIL, Gif-sur-Yvette, Essonne, France. ✉e-mail: ddtuteng@gmail.com; d.dobson@ucl.ac.uk

compaction[15,16]. At shallow mantle depths, magmas are generally less dense than surrounding mantle rocks, leading to melt extraction and, ultimately, volcanic eruption. Several works suggested that a melt-mantle density crossover occurs at depths between 300 and 410 km[14,17,18]. In addition to this density crossover, there is a density trap at 410 km, where the melt density sits between those of olivine and wadsleyite, the dominant minerals of the upper mantle and transition zone, respectively[14,19]. Stable melt layers may form at depths between the density crossover and 410 km through percolation and compaction, whose governing equations are well established in two-phase systems[20,21]. Using such models, the migration rate of 410 melt was estimated to be slow (~0.2 mm/year) compared with the upwelling rate of the ambient mantle (1–10 mm/year), which leads to efficient upwards advection of 410 melt[10]. This process should yield a homogeneous melt layer extending from the 410 km discontinuity to the neutral density location, corresponding to a thickness of 30–100 km[10,20]. This model is here referred to as the "advective thickening (AT) model".

However, the AT model fails to explain the steep lateral variation of layer thickness and the formation of melt doublets. Moreover, the underlying estimation of the rate of melt migration may be problematic because previous studies used the low-pressure properties of dry basaltic melt rather than hydrous ultramafic melt, which is the likely composition of 410 melt. The melt viscosity used was 1 Pa·s, whereas the viscosity of dry basaltic/peridotitic melts under high pressures could be as low as ~10 mPa s[22–25], with water further reducing their viscosity[26–28]. Recent research also challenges the AT model by suggesting that continuous dehydration melting occurs in the ascending mantle due to decreasing water solubility in olivine with decreasing pressure, rather than a single batch melting event atop the 410 km discontinuity[29].

In this study, we present the measurements of the viscosity of hydrous silicate melts at conditions relevant to the bottom of the upper mantle. Based on the viscosity data, we reassess the validity of the AT model and explore the distribution of melt above 410 under the assumption of continuous dehydration melting.

## Results and discussions
### Water effect on the viscosity of silicate melts
We adopted in-situ falling sphere viscometry, a widely recognized technique[22,23,30–33], to measure the viscosity of hydrous peridotitic melt at 14 GPa. Detailed experimental procedures are provided in the Methods section. The elemental composition of the 410 melt was determined in our recent work[34] and was used as the basis for the starting materials, excluding $H_2O$. Its dry composition includes $SiO_2$, $Al_2O_3$, (Mg, Fe)O, and CaO at molar ratios of 29.9, 0.35, 53.6, and 16.2, respectively, which has a much higher Ca content than typical peridotite.

We measured the viscosity of the hydrous peridotitic melts with 15.5, 24.2, and 31.8 mol% $H_2O$ (compositions in Supplementary Table 1) at the temperatures at which they became fully molten. Quench phases in the recovered samples were identified by X-ray diffraction (Supplementary Fig. 8). To determine whether capsules remained sealed against water escape during the experiments, we conducted high-resolution tomography of the recovered samples and measured the volumes of fluid and other quenched phases (Supplementary Fig. 9). The estimated water content in the recovered samples closely matched that of the starting material (Supplementary Tables 1 and 3), confirming that the water was effectively sealed within the capsule during the experiments.

The viscosity of the hydrous peridotitic melt just above the liquidus was found to decrease from 96 (5) to 11.7 (6) mPa·s for melts ranging from 15.5 mol% $H_2O$ to 31.8 mol% $H_2O$ (Fig. 1). These values are slightly higher than those obtained by first-principles simulations[26], but 1–2 orders of magnitude lower than those adopted in previous

compaction simulations[10,20]. The logarithmic viscosity of the 410 melts is approximately linearly related to their $H_2O$ content along their melting curve for $H_2O$ content above 10 mol% (Fig. 1). Compared to the extrapolated viscosity of supercooled dry peridotite melt at the same experimental pressure-temperature conditions, the measured viscosities of the hydrous samples are generally lower, with the exception of the point at ~15 mol% $H_2O$. As illustrated by the higher viscosity of diopside melt compared to enstatite melt (Fig. 1), the presence of Ca can significantly increase melt viscosity. Since the sample composition in this study is much richer in Ca than typical peridotite compositions, the higher viscosity observed at ~15 mol% $H_2O$ can be attributed to the Ca effect. After correcting for the Ca effect, the extrapolated viscosity of dry samples is significantly higher than all the measured viscosities of hydrous samples under the same conditions, which further validates the reliability of our measurements. Previous studies have estimated the $H_2O$ content in the 410 melt to contain more than 15 mol% $H_2O$, with recent estimates exceeding 35 mol%[8,14,26,35]. At the conditions of pressure and temperature relevant to this study, neither Fe content[23,36] nor its valence state[37,38] has a significant effect on the viscosity of 410 melts (see detailed discussion in section Starting material of Methods). Therefore, the viscosity of the 410 melts in the upper mantle should be between 10 and 100 mPa·s.

### The thickness of 410 melt layer in an AT model
Then, we use the newly determined viscosity of the 410 melt to reassess the validity of the AT model. In this model, the migration of the 410 melt is calculated assuming Darcy flow in a porous medium. The mobility of the melt depends strongly on the melt-mantle density contrast, melt viscosity, and melt volume fraction (Eq. (8) in Methods). Larger density contrast, smaller viscosity, and larger melt volume fraction all result in a higher percolation velocity (Fig. 2). The maximum advected melt fraction at any given depth occurs when the downwards percolation velocity is infinitesimally below the upwelling mantle velocity.

With the 410 melts containing more than 15 mol% $H_2O$[8,14,26,35], and the melt fraction in the melt layer limited to be between 0.5 and 1.0 vol%[8,39], a mantle upwelling rate of 10 mm/year could drag up melts with a melt-mantle density contrast of less than 1.3 kg/m³, or even 0.08 kg/m³ for the 410 melts with 35 mol% $H_2O$ (Fig. 2). Larger density contrasts result in melt percolating down faster than the mantle upwells, eliminating advection of melt above 410. A precondition for melt to form LVLs above 410 is that it must be denser than the surrounding mantle. Although the absolute density of the 410 melt remains uncertain, the melt-mantle density contrast decreases almost linearly with decreasing depth above 410 (Supplementary Fig. 10). A melt-mantle density contrast of 0.08–1.3 kg/m³ at a depth of 410 km results in a neutral melt-mantle density position at an elevation of only ~0.08–1.3 km above the 410. This is 1–3 orders of magnitude smaller than what is observed seismically, suggesting that the AT model is incapable of making a thick low-velocity layer.

### Formation of melt doublet in a continuous dehydration melting model
Now we consider the impact of continuous dehydration melting (CDM) in the upwelling upper mantle. In this scenario, the melt distribution is controlled by the balance between rates of melt generation, percolation, and drainage. Above the neutral density position (NDP), melt migrates up to the surface quickly because it is buoyant. At intermediate depths between 410 and the NDP, the melt can segregate upwards or downwards depending on the balance between advection and percolation. We simulated the evolution of melt distribution in this mantle region using a 1D finite element method. However, a simple model of Darcy flow in porous media is insufficient. Our simulation accounts for factors of plastic mantle flow (compaction), gravitational segregation, dehydration melting, and capillary action in the melt

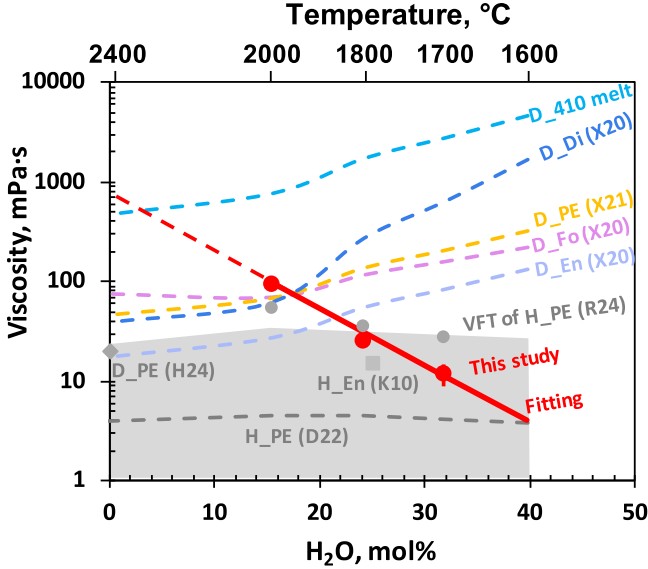

**Fig. 1 | Viscosity of hydrous peridotitic melt along its melting curve as a function of water content.** Red circles represent experimental data from this study, with the red solid line representing a linear fit of the logarithmic viscosity as a function of water content. The red dashed line indicates the linear extrapolation to dry conditions. Gray circles indicate predictions based on the Vogel-Fulcher-Tammann (VFT) model of hydrous peridotite from Russel et al. [59]. To isolate the effects of compositions and temperature, extrapolated viscosity of dry super-cooled silicate liquids—including diopside [23], enstatite [23], forsterite[23], peridotite[22] and 410 melt compositions[34]—are shown as colored dashed lines, calculated under the same pressure-temperature conditions as the experiments in this study. Since the water content in the dry super-cooled silicate liquids is zero, the lower horizontal axis is invalid for them. The viscosity of dry 410 melt was derived using an Arrhenius equation for forsterite melt with Ca-corrected activation enthalpy (see Methods for details). The gray dashed line with an error envelope represents viscosity estimates for hydrous peridotitic melt based on first-principles (FP) calculation[26]. The gray square marks the viscosity of liquid hydrous enstatite[28], while the gray diamond indicates the viscosity of dry peridotite derived from FP calculations[24]. D_Di dry diopside, D_En dry enstatite, D_Fo dry forsterite, D_PE dry peridotite, D_410 melt dry 410 melt composition from Xie et al. submitted[34], H_En hydrous enstatite, H_PE hydrous peridotite. Source data are provided as a Source Data file.

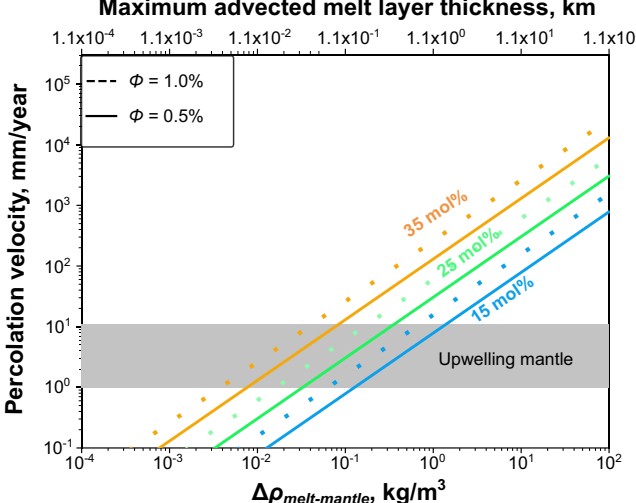

**Fig. 2 | The percolation velocity as a function of melt-mantle density contrast ($\Delta\rho$) for various melt fractions ($\Phi$) and viscosities.** The orange, green, and blue lines indicate the percolation velocity calculated with the viscosity corresponding to the 410 melt with 35, 25, and 15 mol% $H_2O$, respectively. Solid and dotted lines correspond to 1.0 and 0.5 vol% of melt, respectively. The gray region represents the typical range of mantle upwelling rate (1–10 mm/year).

distribution model. The melt evolution is governed by equations of mass conservation and the momentum equation describing the matrix-melt interaction (for simulation details, see Methods). The parameters were set as follows: (i) initial melt distribution is homogeneous at 1 ppm by volume; (ii) melt viscosity with a conservative value of 20 mPa·s, corresponding to a water content of ~25 mol% $H_2O$; (iii) melt-mantle density contrast of 0–300 kg/m³ at 410 km depth, with the latter value corresponding to the maximum possible value of mantle's density jump at the 410 km discontinuity; (iv) density contrast decreasing linearly with depth (Supplementary Fig. 10), until it becomes zero at the NDP; (v) other parameters such as the mantle velocity are listed in Supplementary Table 4.

A result of our simulation is the appearance of two distinct melt layers: one atop the density trap at 410 km depth, and the other slightly below the depth of NDP (Fig. 3). The former layer forms due to the density trap and efficient melt segregation caused by a relatively high melt-mantle density contrast. The latter layer emerges due to equal, but opposite, velocities for mantle advection and melt percolation as the density contrast diminishes. These two layers can either stabilize as distinct layers (Fig. 3a) or merge into a single layer over time (Fig. 3b), depending on the conditions. This melt doublet system develops and stabilizes within 0.5 Ma, with layer thicknesses of tens of kilometers, comparable to the observed LVLs. As a globally occurring process, the

CDM model explains the worldwide distribution of melt layers atop the 410 km discontinuity.

To further understand the dynamics of the melt doublet system in the CDM model, we analyzed how various parameters affect the layer pattern and thickness in the melt doublet system under typical mantle conditions. The primary factors include the depth of the NDP, the melt-mantle density contrast at 410 km, the mantle viscosity and the mantle upwelling rate (Fig. 4). A shallower NDP results in a larger distance between the two layers, promoting the stability of a two-layer pattern (Fig. 4a). A larger melt-mantle density contrast enhances melt percolation, also favoring a stable pattern of two distinct layers (Fig. 4b). Similarly, a lower mantle viscosity promotes plastic flow and compaction of solid material, which further favors a stable two-layer pattern (Fig. 4c). Conversely, a higher rate of mantle upwelling increases the melt generation flux and decreases the drainage rate, resulting in a single melt layer pattern (Fig. 4d). In comparison, the amount of melt generated upon mantle upwelling (due to progressive dehydration of the solid residue) and the surface tension have minor effects (Supplementary Fig. 13). Notably, both configurations—two distinct layers or a single layer—can form under typical mantle conditions. The layer thickness in the single-layer regime corresponds approximately to the depth interval between 410 km and the NDP. The thickness of layers in the two-layer regime is much smaller, varying from 0 to ~40 km depending on conditions. A sharp change in layer thickness occurs when these two regimes transition between each other, which could explain the steep lateral variations of LVLs observed atop the 410.

To evaluate the robustness of the CDM model, we explored the transition between the AT and CDM models by investigating the impact of the thickness of the melt generation layer on melt distribution. As shown in Supplementary Fig. 14a, a CDM model converts to an AT model when the thickness of the melt generation layer is less than ~2/3 (~60 km) of the depth interval between 410 and the NDP. Using the 1D finite element method, we also investigated the effect of mantle compaction on the AT model. The advected length of melt increases with decreasing mantle viscosity (Supplementary Fig. 14b). However, the dragging length is less than 3 km within the possible viscosity range of the upper mantle, supporting the result of the Darcy flow calculation (Supplementary Fig. 14b).

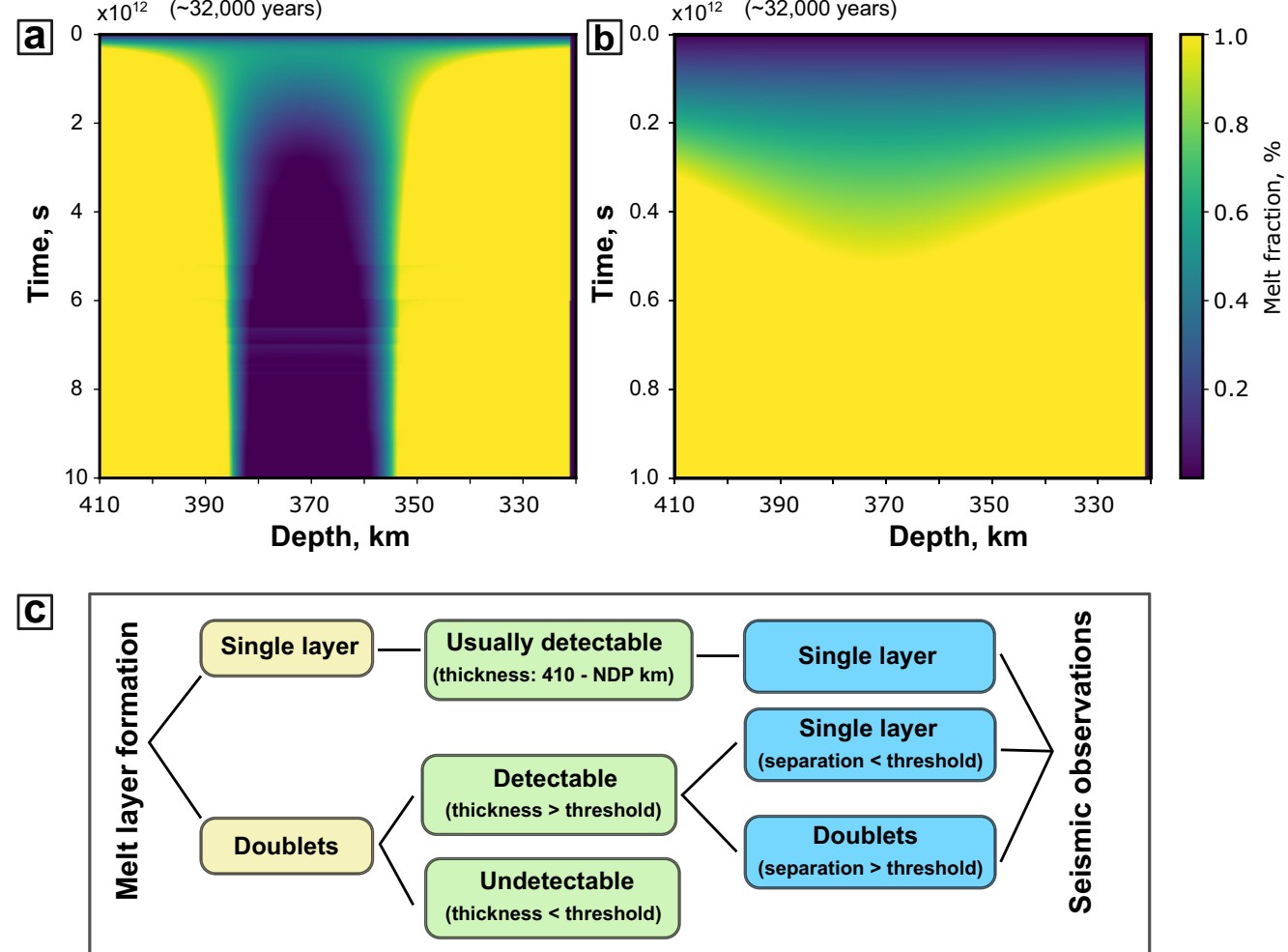

**Fig. 3 | Representative time evolution patterns of melt distribution in a continuous dehydration melting model. a** Example of melt distribution evolving towards two distinct melt layers for a melt-mantle density contrast of 100 kg/m³. In this simulation, thicknesses of the lower and upper melt layer are ~25 and ~35 km, respectively. **b** Example of melt distribution evolving toward a single melt layer for a melt-mantle density contrast of 50 kg/m³. **c** Conceptual diagram illustrating the detectability of melt layers using the receiver function method. Threshold is the seismic detection limit for the melt layer/separation thickness, which depends on the wave frequency.

## Implications for the seismic observations

Finally, we compare the CMD model results with seismic observations. Detecting melt layers seismically requires a thickness threshold that depends on the wave frequency (Supplementary Fig. 15). Considering the noise level in real seismic data, the threshold should be thicker[2]. If observable, the receiver functions show the top of the low-velocity layer, with 1 negative seismic phase in the case of a single-layer model (Supplementary Fig. 15a, b) and 2 negative phases for the melt doublet model (Supplementary Fig. 15c, d). Figure 3c outlines the conditions under which melt layers can be detected using receiver functions. A single, merged melt layer is generally easier to identify due to its larger thickness, while two distinct melt layers are only detectable if their individual thicknesses exceed the threshold. Additionally, a separation, the distance between the tops of two layers, threshold is necessary for seismic distinction (Supplementary Fig. 15c, d). The bottom layer may be undetectable, and only the top layer is detectable when the separation is small, but larger separations allow for the clear identification of two distinct melt layers (Supplementary Fig. 15c, d). In previous global studies, the threshold for detection is ~20 km. Approximately 60% of seismic stations detect LVLs, some of which are possibly doublets. Observation of a single LVL represents either a single thick melt layer or a melt doublet where the separation between the two layers is below the seismic resolution. An observed LVL

doublet indicates a melt doublet. The coexistence of LVL single and double layers infers that the typical mantle conditions are near the transition condition between a single merged melt layer and melt doublet, where the layer thickness varies dramatically, explaining the steep lateral variation of LVL thickness. Conversely, the 40% of stations that do not detect LVLs likely correspond to regions where two distinct thin melt layers exist, both below the detection threshold (Fig. 4). Note that our study applies to regions of mantle upwelling, and other explanations are applicable to melts observed around subducting slabs.

One prediction of the CDM model is the conversion between a single merged melt layer and a melt doublet due to the changing of the primary factors, such as the mantle upwelling rate or melt density, atop 410. Interestingly, this phenomenon is clearly shown in regional studies of Afar Triple Junction, which used high-quality and relatively high-frequency receiver functions. Underneath the Afar Triple Junction, a single LVL has been observed at the plume center while it converted to two LVL layers ~100 km away, at the plume edges (Supplementary Fig. 2)[6]. The receiver functions used in the Afar study adopted a Gaussian pulse width of ~2 s, which theoretically can detect melt layers as thin as ~10 km (Supplementary Fig. 15a). The thinnest LVL in Afar are ~20 km thick (Supplementary Fig. 2), which requires a separation of ~50 km to distinguish two close layers (Supplementary

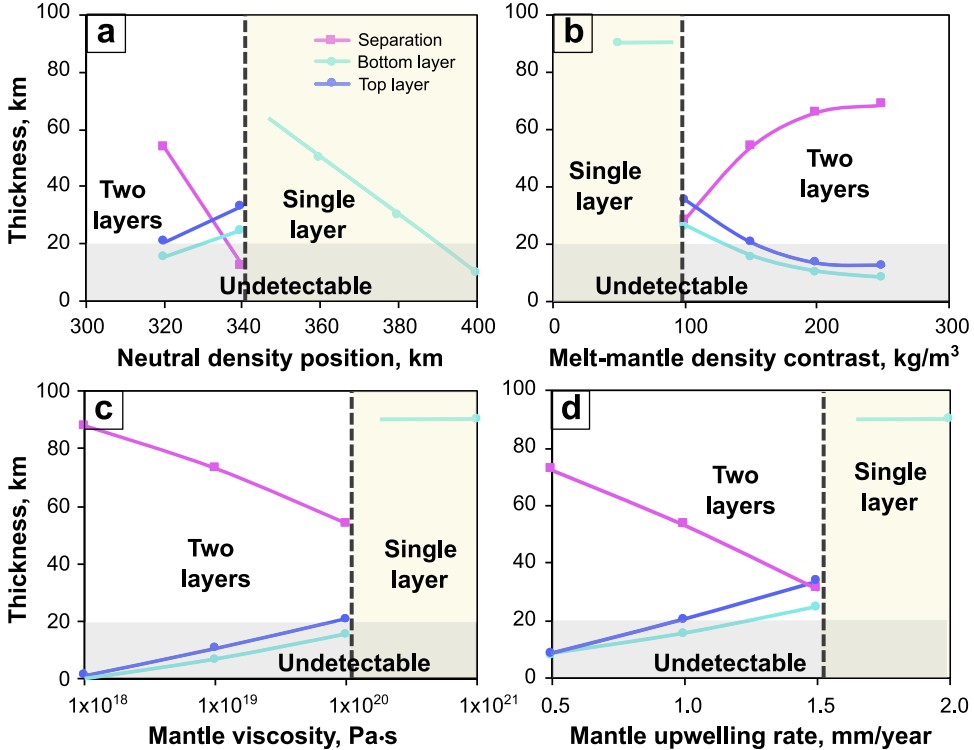

**Fig. 4 | Major factors that affect the number, thickness, and layers' separation of melt layers in a continuous dehydration melting model.** Effects of (**a**) depth of the neutral density position, (**b**) melt-mantle density contrast at 410 km, (**c**) mantle viscosity and (**d**) mantle upwelling rate. Other parameters are listed in Fig. 15c). The separations of two LVL layers at the plume edge exceed Supplementary Table 4. The yellow regions indicate conditions where a single layer forms, while the gray regions represent layers that are undetectable in a global analysis. Source data are provided as a Source Data file.

Fig. 15c). The separations of two LVL layers at the plume edge exceed 60 km (Supplementary Fig. 2), confirming validity of the conversion between single melt layer and melt doublet in Afar and, thus, supporting our CMD model. Given that upper mantle viscosity increases toward the plume edges, favoring a single merged melt layer rather than the observed layer doublet, and considering that melt density remains relatively constant within the same location, this conversion is likely driven by higher upwelling rates at the plume centers and lower rates at the edges (Supplementary Fig. 16).

In conclusion, the newly determined viscosity of the water-bearing 410-melt is remarkably low, favoring rapid segregation of melt in the upper mantle above 410. At these low viscosities, the simple AT model can't produce melt layers of significant thickness above 410. Instead, continuous dehydration melting in the upwelling mantle above 410 can deliver melt to two dynamically stable melt regions (the density trap above 410 and the zone below the NDP), forming thick melt layers. This model of a melt doublet system allows the formation of either (i) a pattern of two distinct melt layers, at the density trap and beneath the NDP, or (ii) a single thick melt layer, if the two melt layers merge. The melt doublet system explains many global and regional seismic observations of low-velocity zones above 410. We conclude that a melt doublet system forms at the base of the upper mantle due to the continuous dehydration melting in regions of mantle upwelling.

## Methods
### Experimental methods
**Starting material.** The chemical composition of 410 melt is likely to be ultramafic and water-bearing[13,14,40]. Although the water content in the 410 melt varies among different studies, the ratios of the elements, except $H_2O$, are consistent[13,14,41]. Although Fe content has a significant effect on the viscosity of silicate melt at low pressures, it should have little effect on the viscosity of 410 melt at pressures above 10 GPa[23,36]. An experimental study[38] demonstrated that the valence of Fe has little

effect on the viscosity of silicate melts at moderate to low values of $Fe^{3+}$/total Fe (<0.4). The activity of $H_2O$ can influence oxygen fugacity ($fO_2$) through its dissociation reaction into hydrogen and oxygen gases, which in turn affects the valence state of Fe in Fe-bearing silicate melts[37]. In the upper mantle, however, the $f(O_2)$ is controlled by the quartz-fayalite-magnetite (QFM) system because the quantity of hydrous silicate melt generated under these conditions is tiny. The upper mantle $f(O_2)$ is at or lower than the QFM buffer and gets more reducing with depth[42]. Estimates suggest that even at subduction zones, $f(O_2)$ is 1–2 log units below QFM by 200 km depth[42]. The $Fe^{3+}$/total Fe ratio in silicate melts is -0.15 in a QFM buffered upper mantle[37], which is well below the threshold of 0.4. Therefore, the viscosity of Fe-free 410 melt is a good approximation of that of 410 melt. In light of this, and in order to avoid the density change due to Fe loss to the Pt capsule lid, Fe-free starting materials were used. To maintain a constant (Mg + Fe + Ca)/Si ratio in the melt, its FeO content was replaced by MgO. Reagent grade powders ($SiO_2$, $Al_2O_3$, MgO, $Mg(OH)_2$, $Al(OH)_3$, and $Ca(OH)_2$) were mixed in an appropriate ratio (Supplementary Table 1) and hand-ground using an agate mortar and pestle. Water was added in the form of $Ca(OH)_2$, $Mg(OH)_2$, or $Al(OH)_3$ to produce mixtures with target water contents of 15.5, 24.2, and 31.8 mol%, corresponding to 6.28, 10.54, and 14.6 wt% for the Fe-free starting materials.

**High-pressure, high-temperature experiments.** The viscosity measurements were conducted at SPring-8, Japan. High pressures were generated using a 1500 ton Kawai-type multi-anvil press, SPEED-1500, installed in the front hutch of beamline BL04B1. Cubic anvils made of tungsten carbide with 8 mm truncated edge lengths were adopted as second-stage anvils.

Supplementary Fig. 3 shows the cell assembly parallel to the X-ray incidence, slightly modified from the cell assembly developed in ref. 23. A Cr-doped MgO (5 wt% $Cr_2O_3$) octahedron with a 14 mm edge length was used as a pressure medium. A boron-doped diamond (BDD)

tube, which is X-ray-transparent for radiography, is used as the heater to melt the sample[22,23,43,44]. A $W_{75}Re_{25}$-$W_{97}Re_3$ thermocouple with 0.05 mm diameter was used to monitor the temperature and was insulated from the heater with two $Al_2O_3$ tubes. Diamond single-crystal tubes were used as capsules, with the two ends of the capsules sealed with Pt metal disks. The testing experiments on capsule sealing were conducted at beamline Psiché, SOLEIL, France. Pt and Re spheres (50–100 µm diameter) were tested as falling spheres. Because Pt spheres are too soft, they deform during compression, and so Re spheres were used in the viscosity measuring experiments instead. A mixture of nanodiamond powder (~5 nm diameter) and Pt powder (1 µm diameter, 50 wt%) was placed into the top end of the diamond tube as a pressure marker. Nanodiamond powder was used to prevent the grain growth of Pt. Pressures were calibrated against the Pt pressure scale[45].

Diamond single-crystal tubes were fabricated from CVD-diamond single crystal chips, purchased from Chenguang Machinery & Electric Equipment Co, Ltd (China). These tubes were shaped using either an Oxford Instruments A-series laser milling machine at the University of Bristol (UK) with a wavelength of 532 nm and a pulse-width of 40 ns operating at pulse frequencies of 5–15 kHz and output power of 50–100% (maximum average power is 11 W at 40 kHz) or an infrared laser at the Earth & Planets Laboratory (US). The metallic spheres were prepared by fusing Re and Pt wires[23]. The Re wires were immersed in liquid nitrogen to prevent oxidization and to enhance the quenching rate. The sphere size at ambient pressure was measured by both SEM and X-ray radiography to obtain a calibration line (Supplementary Fig. 4). The sphere size at high pressure was measured in-situ radiographically and then calibrated using the obtained calibration line.

A whitebeam was used for both pressure determination and imaging of the falling sphere. Energy-dispersive powder X-ray diffraction was collected using a Ge solid-state detector. Behind the press, a phosphor plate was positioned to convert the X-ray intensity to visible light. A fast camera (C11440, >1000 f/s) with a 75 mm objective lens was focused onto the phosphor to capture the images of falling spheres.

The cell assemblies were first compressed to 780 ton at room temperature, and the pressure was measured at the target load. Subsequently, temperature was increased slowly to ~1000 °C, and the pressure was measured again. Then, the fast camera was set to take photos at a rate of 1000 f/s, and the sample was heated to the target temperature within 0.1 s. The electrical power required to reach the target temperature was estimated from the power-temperature relationship obtained before fast heating. After observing the sphere fall, the samples were quenched by cutting off the electrical power, decompressed, and recovered from the press. The recovered samples were analyzed by X-ray diffraction and X-ray tomography.

**Temperature and pressure estimations during a sphere fall.** The temperature before fast heating was recorded by a thermocouple. The target temperatures during fast heating were set to lie along the liquidus curves of the forsterite-water system[46], which has a $(Mg + Fe + Ca)/Si$ ratio close to the 410 melt. The temperature during sphere fall was estimated by the power-temperature relation obtained before fast heating and almost falls on the liquidus of the forsterite-water system (Supplementary Fig. 5).

Supplementary Fig. 6a summarizes the pressures inside the diamond capsules of various runs during compression at room temperature. The pressure increased slowly (~0.64 GPa per 100 tons) with increasing loads due to the incompressibility of the diamond capsule. The pressures at the target load (780 tons) range from 3 to 5.3 GPa, with an average of $4.9 ± 0.9$ GPa. Upon heating, thermal pressure within a diamond tube is expected to be significant[22,23,43]. Supplementary Fig. 6b summarizes the thermal pressures inside the diamond capsules of various heating runs performed at 780 tons. The thermal

pressure is found to be 0.56 GPa per 100 °C. The scattering of pressure slightly decreases with increasing temperature, with the standard deviation of pressures decreasing from 0.9 at room temperature to 0.7 at 1000 °C.

The sphere fell very fast, transiting the sample within 100 ms. The experiments were then quenched, directly after the sphere fell, to prevent destruction of the sample at the very high temperatures involved. Pressure during fast heating was therefore estimated through the pressure-temperature relation obtained at lower temperatures. The temperatures and pressures at which the spheres fell are summarized in Supplementary Table 2.

**Viscosity calculation and error analysis.** The melt viscosity ($\eta$) was obtained from the terminal velocity based on Stokes' law:

$$\eta = \frac{2gr_s^2(\rho_s - \rho_m)W}{9\nu_s E} \qquad (1)$$

$$W = 1 - 2.104\left(\frac{r_s}{r_c}\right) + 2.09\frac{r_s^3}{r_c} - 0.95\frac{r_s^5}{r_c} \qquad (2)$$

$$E = 1 + 3.3\left(\frac{r_s}{h_c}\right) \qquad (3)$$

where $\nu_s$, $r_s$, $\rho_s$, $\rho_m$, and $g$ are the terminal velocity, sphere radius, sphere density, melt density, and gravity acceleration, respectively. The parameters $W$ and $E$ are for correction due to the finite dimensions of the sample chamber with a radius $r_c$ and height $h_c$[47]. The density of Re sphere and 410 melt were calculated using the equation of state of Re[48] and eq. of state of hydrous silicate melt[26]. The terminal velocity was obtained from the sphere position-time or velocity-time diagram (see e.g., Supplementary Fig. 7). The propagation of experimental uncertainties was evaluated using a Monte Carlo method[23].

**Phase identification and tomography analysis of recovered samples.** Phase identification in the recovered sample (S3642) was conducted at SPring-8 using a monochromatic beam of 60.6583 keV. To achieve a good signal-to-noise ratio, the MgO pressure medium outside the heater was mechanically removed. However, to prevent damage to the sample, the heater, the MgO sleeve inside, and the diamond capsule were retained during the X-ray diffraction measurement. A total of four phases—phase A ($Mg_7Si_2O_8(OH)_2$)[49,50], diopside, breyite ($CaSiO_3$), and MgO—were identified (Supplementary Fig. 8). Since the MgO signal likely originates from the MgO sleeve, the recovered sample consists of phase A, diopside, and breyite.

To estimate the phase volumes and identify possible voids in the recovered samples, tomography was performed on samples from Runs S3640, S3641, and S3645 using the EasyTom 150–160 tomograph from RXSolutions installed at Laboratoire Magmas et Volcans, France. Similar to the X-ray diffraction measurements, the MgO pressure medium outside the heater was removed, and only the components within the heater were subjected to tomography. The voxel size was 0.216 µm³.

Supplementary Fig. 9a provides an example of a tomographic model from Run S3641, while Supplementary Fig. 9b displays a slice of the X-ray absorption coefficient contrast image. In these images, higher brightness indicates a higher X-ray absorption coefficient. A total of three brightness groups, excluding diamond, were clearly identified. Since voids are more transparent than diamond, regions within the diamond capsule that exhibit a similar or lower X-ray absorption coefficient than diamond were classified as voids. Numerous irregularly shaped voids were detected in the recovered samples. Given that diopside/breyite has a higher X-ray absorption coefficient than phase A, the phase with medium brightness was assigned to phase

A, while the brightest phase was attributed to diopside/breyite. A gray value histogram of the sub-sample volume was plotted to determine the thresholds for isolating phases (Supplementary Fig. 9c). The two valleys between the three peaks were defined as the thresholds. A 3D model of the region of interest (ROI) was reconstructed for each phase (e.g., Supplementary Fig. 9d), and the voxel numbers of the 3D model were counted (Supplementary Table 3).

The water content in the recovered samples was estimated based on the calculated volume content of the different phases. The voids were assumed to have been composed of Ice VII, with an ambient pressure density of $0.92\,g/cm^3$. To estimate the water content in weight percent, densities of 3.0 and $3.25\,g/cm^3$ were used for phase A and diopside/breyite, respectively. The total $H_2O$ content was calculated as the sum of $H_2O$ present in phase A and Ice VII (Supplementary Table 3). We identify possible sources of uncertainties, however. Since the pressure may not be completely released in the diamond capsule, the voids likely represent the volume of Ice VII under pressure. Thus, the estimated Ice VII content is a minimum amount. Secondly, breyite presents a slightly lower density ($\sim3.0\,g/cm^3$) than diopside ($\sim3.4\,g/cm^3$). In the brightest regions of the sample, we do not know precisely the phase fraction of these two phases. This could lead to an overestimation of the diopside/breyite content in wt%, and consequently, an underestimation of the Ice VII and phase A content. As a result, the $H_2O$ content presented in this study is considered conservative.

## Estimation of viscosity of dry 410 melt at 14 GPa

The Arrhenius equations of viscosity of forsterite (Fo), enstatite (En), and diopside (Di) melts have been parametrized in ref. [23]. These equations show that both the (Mg + Fe + Ca)/Si ratio and Ca content significantly affect the activation enthalpy. The 410 melt has a (Mg + Fe + Ca)/Si ratio similar to that of Fo. Thus, the Arrhenius equation for the viscosity of Fo melts can serve as a rough approximation for that of dry 410 melt after accounting for the Ca effect on activation enthalpy.

Assuming a linear effect of Ca on the activation enthalpy, the activation enthalpy of the 410 melt can be approximated by the following equation:

$$\Delta H_{410\,melt} = \Delta H_{Fo} + R_{Ca}\frac{(\Delta H_{Di} - \Delta H_{En})}{0.5} \tag{4}$$

where $\Delta H$ represents the activation enthalpy, and $R_{Ca}$ = Ca/(Ca + Mg + Fe) is the mole fraction of Ca, which is 0.23 for 410 melt. Using this equation, the activation enthalpy of 410 melt is calculated to be 160 kJ/mol. With the pre-exponential factor for Fo melt ($1.72 \times 10^{-4}$), the Arrhenius equation for the viscosity of 410 melt ($\eta$) becomes:

$$\eta = 1.72 \times 10^{-4}e^{\frac{160000}{RT}} \tag{5}$$

where $R$ is the gas constant, and $T$ is the temperature in K.

The estimated viscosity of the 410 melt is in the same order as that obtained through linear extrapolation of the logarithmic viscosity (Fig. 1), suggesting that Ca may play a significant role in causing the large viscosity difference between 410 melt and Fo/peridotite melt.

## The linear relationship between melt-mantle density contrast and depth

Besides viscosity, the melt-mantle density contrast is another key parameter to understand the migration and storage of melt atop the 410. While the density of the mantle is well constrained by the preliminary reference Earth model (PREM)[19], the absolute density of hydrous silicate melt remains debated[14,26,51] due to uncertainties in its equation of state and composition, particularly with respect to Fe and water content. Despite these uncertainties, the melt-mantle density contrast is roughly linearly correlated with depth between 300 and

410 km (Supplementary Fig. 10). For our simulations, we treated melt density as a variable at the 410-km depth, ranging from 0 to $300\,kg/m^3$ higher than the mantle, with a linear decrease to zero at the NDP.

## Percolation velocity of the 410 melt

Buoyancy is the driving force for the possible 410-melt segregation in the upwelling upper mantle. A balance between the buoyancy force and viscous drag leads to the percolation velocity ($\Delta\mu$), which is the difference between mantle upwelling ($v_m$) and melt sedimentation ($v_f$) velocities:

$$\Delta\mu = \frac{kg\Delta\rho}{\eta\phi} \tag{6}$$

where $k$ ($m^2$) is the permeability of ambient mantle, $g$ is the gravitational acceleration, $\Delta\rho$ ($kg/m^3$) is the melt-mantle density contrast, $\eta$ (Pa·s) is the melt viscosity, $\phi$ is melt fraction.

For porosity less than 2%, the permeability can be well approximated by refs. [52,53]:

$$k = \frac{d^2\varnothing^2}{1600} \tag{7}$$

where $d$ is the typical mantle grain size $\sim1\,mm$. Substituting Eq. (7) in (6), we obtain:

$$\Delta\mu = \frac{d^2g\Delta\rho\phi}{1600\eta} \tag{8}$$

The average upwelling velocity is $\sim1$–$10\,mm/year$. Melts denser than the mantle are dynamically static if they percolate downward with the same velocity.

## Method for percolation simulation

**Governing equations.** Mantle flow, gravitational segregation, dehydration melting, and capillary action on grain boundaries are the primary factors influencing melt migration and distribution in a partially molten aggregate. We consider the percolation of melt in a one-dimensional column in the steady state. Both the solid matrix and the melt are viscous fluids with viscosities $\mu_m$ and $\mu_f$. They have densities $\rho_m$ and $\rho_f$, respectively. Following the analysis and equations outlined by Ricard et al. and Hier-Majumder et al. for one-dimensional forced compaction, the equation for mass conservation of the matrix is given by refs. [20,54]:

$$\frac{\partial\phi}{\partial t} = \frac{\partial}{\partial z}\left((1-\phi)v_m\right) + f(1-\phi)v_m \tag{9}$$

where $\phi$ is the melt fraction, which varies spatially and temporally, and $f$ is the degree of dehydration melting.

The momentum equation, describing the interaction between matrix and melt, is:

$$0 = (1-\phi)\chi''\frac{\partial\phi}{\partial z} + \frac{\partial}{\partial z}\left(\mu_m\left(\frac{K_0}{\phi} + \frac{4}{3}\right)(1-\phi)\frac{\partial v_m}{\partial z}\right) - (1-\phi)\Delta\rho g - \frac{cv_m}{\phi^2}, \tag{10}$$

where $c = \frac{\mu_f\phi^2}{k}$, $\chi$ represents the total interfacial tension per unit volume, $\chi''$ is the second derivative of $\chi$. Permeability $k$ follows Eq. (8).

Since the melt fraction in 410 LVL is less than the disaggregation melt fraction (0.21)[20] and the mineral-melt dihedral angle $\theta$ is very small, $\chi''$ can be expressed as[20]:

$$\chi'' = \frac{1}{d}\left[(-3.3908\sigma_{mf} + 2.1587\sigma_{mm})\phi^{-\frac{3}{2}}\right] \tag{11}$$

where $\sigma_{mf}$ of $0.5 \pm 0.2$ (J/m$^2$) and $\sigma_{mm}$ of $1.6 \pm 0.3$ (J/m$^2$)[55] are the interfacial energy of mineral-mineral and mineral-melt, respectively.

**Finite element method for solution.** To solve the time-dependent partial differential Eqs. (9) and (10), a finite element method is employed. Eq. (9) is discretized as follows:

$$\phi_n = \phi_{n-1} + dt \frac{\partial}{\partial z}\left((1-\phi_n)v_m\right) + f(1-\phi_n)v_m dt. \tag{12}$$

Rearranging, we obtain:

$$(1 + dt\nabla v_m + fv_m dt)\phi_n + dtv_m\nabla\phi_n = \phi_{n-1} + dt\nabla v_m + fv_m dt, \tag{13}$$

where $\nabla$ is the gradient operator. Setting $\phi_n$ as the trial function and $v_\phi$ as the test function, the variational form of Eq. (13) is:

$$\int [(1 + dt^*\nabla v_m + dt^*fv_m)\phi_n + dt^*v_m\nabla\phi_n]v_\phi dz$$

$$= \int [\phi_{n-1} + dt^*\nabla v_m + dt^*fv_m]v_\phi dz \tag{14}$$

To solve Eq. (14), firstly, we need to obtain $v_m$ as a function of depth $z$ and time. At the neutral density position (NDP) and 410 km, the $v_m$ is a constant and, thus, $\nabla v_m$ is 0. Setting $v_m$ as the trial function and $v_{v_m}$ as the test function, the variational form of Eq. (10) is:

$$\int \mu_m\left(\frac{K_0}{\phi_{n-1}} + \frac{4}{3}\right)(1-\phi_{n-1})\nabla v_m \cdot \nabla v_{v_m} dz - \int \frac{cv_m}{\phi_{n-1}^2}v_{v_m} dz$$

$$= \int [(1-\phi_{n-1})\Delta\rho g]v_{v_m} dz - \int [(1-\phi_{n-1})\chi''\nabla\phi_{n-1}]v_{v_m} dz \tag{15}$$

We mesh the depth interval between NDP and 410 km into nz patches and track the evolution of melt distribution from an almost dry mantle (with 1 ppm melt homogeneously distributed in the volume). A constant melt generation rate was adopted. The top and bottom 1% of the depth interval of the thickness of mantle considered here were set as the top and bottom boundary, respectively. At each time step, $v_m$ is first solved using Eq. (15) and then $\phi_n$ is solved using Eq. (14). The partial equations were solved using the python function library "fenics"[56].

**Mesh and boundary conditions.** Supplementary Fig. 11 shows examples of simulations using different mesh sizes for the time and spatial domains. Both parameters happen to have little effect on the simulated melt distribution pattern when using a total temporal (nt) and spatial (nz) mesh number larger than 100 (Supplementary Fig. 11). Considering the computational time required, we arbitrarily select the total temporal and spatial mesh number as 1000 and 10,000, respectively.

We tested 3 types of boundary conditions of the bottom boundary, i.e., constant value, exponential decay function, Gaussian decay function. The types of boundary conditions have little effect on the formation of melt layers during the melt percolation (Supplementary Fig. 12). Because the top boundary continuously loses melt upwards and the bottom boundary also loses melt through recycling back to the transition zone, a Gaussian decay function was adopted. Furthermore, a maximum limit was set as 1% for the melt fraction in melt layers to account for lateral flow.

**Method for receiver function synthesis**
We used the traditional reflectivity method to synthesize the receiver functions of single-layer or melt doublet models using the "Raysum" software[57]. The seismic wave velocities at various depths were derived from the IASP Earth velocity model[58] through interpolation, with melt layers having a velocity 5% lower than the IASP91 Earth

velocity model. We generate receiver functions with two widths of Gaussian pulse (2 and 4 s) to investigate the effect of wave frequency on the detection resolution of LVLs. In general, the detection resolution increases when using higher frequency receiver functions (Supplementary Fig. 15). For most receiver functions studies of the transition zone, a Gaussian pulse width of ~2 s (corresponding to a Gaussian filter width of 0.625) represents an upper bound, suggesting these models show the minimum thickness observable in a best-case scenario. For a Gaussian pulse width of ~2 s, a single layer must be >10 km to be observable, and two melt layers with 20 km thickness must have a minimum of 50 km distance between them to be observed as two distinct phases. With a pulse width of ~4 s, a single melt layer must be thicker than 20 km, and a pair of 20 km-thick layers requires a separation of at least 80 km to be resolved as distinct.

## Data availability
The authors declare that the data supporting the findings of this study are available in the paper or in supplementary materials. The data generated in this study have been deposited in the zenodo under accession code https://doi.org/10.5281/zenodo.14871120. Source data are provided with this paper.

## Code availability
The images were analyzed using public software Fiji, which is an open-source image processing package based on ImageJ. The finite element simulation was conducted with an open-source Python function library "fenics". The receiver functions were synthesized using the open-source "Raysum" software.

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

## Acknowledgements

The viscosity and X-ray diffraction measurements were conducted at BL04B1, SPring-8, Japan under the proposal of 2023A1109 and 2024A1175. The diamond sealing technique was developed at Psiché, SOLEIL, France under the proposal of 20230084, 20220234, 20211568, and 20201203. L.X. was supported by RCUK grants (NE/X009807 and NE/T006617) to D.D. L.X. received support for conducting the tomography of recovered sample under the 2024 Visitor Program of ClerVolc (contribution number 686) and supports for beamtime preparation under the 2023 Joint Use/Research Center for "Earth and Planetary Materials Sciences", Institute for Planetary Materials (IPM), Okayama University, Japan. OTL would like to acknowledge support from the Royal Society in the form of a University Research Fellowship (UF150057). The projects received supports from JSPS KAKENHI (Grant Number: JP21H04996) to T.Y., and RCUK grants (NE/X009807 and NE/T006617) to D.D.

## Author contributions

L.X. designed the project. L.X., D.A., and D.D. constructed the concept model. L.X. performed the viscosity measurements with D.D., T.Y., F.X., B.Z., S.K., N.T., and Y.H. L.X. laser cut the diamond capsule with Y.F. and O.L. L.X., D.A., L.H., and N.G. developed the technique of sealing melt in the diamond capsule. S.F., D.A., and L.X. conducted the tomography imaging of recovered samples. L.X. did the data analysis. L.X. conducted the percolation velocity calculation in a Dacy flow model and the 1D simulation. C.H. and J.H. did the receiver function synthesize. L.X. wrote the first draft of the manuscript. L.X., D.A., T.Y., D.D., C.H., J.H., F.X., O.L., Y.F., and B.Z. contributed to the discussion of the results and revision of the manuscript.

## Competing interests

The authors declare no competing interests.
