## [Peer Review File · Nature Communications]

Low Melt Viscosity Enables Melt Doublets Above The 410-km Discontinuity

Corresponding Author: Dr Longjian Xie

Editorial Note: Figures on page 9 and 10 of this Peer Review File have been redacted as indicated to remove third-party material where no permission to publish could be obtained.

Version 0:

Reviewer comments:

Reviewer #1

(Remarks to the Author)

I have now reviewed the manuscript "Low Melt Viscosity Enables Melt Doublets Above The 410-km Discontinuity" by Xie et al., submitted to Nature Communications. This study address the origin of the seismic discontinuities at 300-410 km depths in the Earth's mantle, for which partial melts may provide a viable explanation. In this work, the authors performed extremely challenging high-pressure experiments to measure the viscosity of hydrous-bearing iron-free silicate melts, and then used the results together with other available physical information on the properties of mantle melts to model melt segregation above the 410 km depth.

Overall, this is a very thorough study and I applaud the authors for their effort. Yet, my background is in physical properties of glasses and melts, and I am unable to judge the technical quality of the numerical simulations performed here (to me they seem sufficiently comprehensive). I hope another reviewer will provide their expert opinion on the quality of numerical simulations. The experimental part, however, has issues and can be improved. After these issues are resolved, the paper may become suitable for publication in Nature Communications.

My main concern lies with the reliability of the presented viscosity measurements, which is difficult to assess by comparing to reference materials with constrained viscosity at similar conditions. The ability of the used experimental technique to reproduce literature data is not demonstrated in the present manuscript. A corresponding (i.e., Fe-free) dry ultramafic melt was not probed here, although this could have allowed an independent check on the reliability of the viscosity measurements. Below, I examine the presented results by comparing them to authors' own data on the viscosity of an Fe-bearing dry peridotitic melt (Xie et al., GRL, 2021) but also to the new model of Russell et al., 2024 (not cited in this work, but certainly relevant): <https://www.sciencedirect.com/science/article/pii/S0012821X24003327>

One problem I see is that the previous study from the same group and using the same method (Xie et al., GRL, 2021) suggests that the viscosity of an Fe-bearing and dry peridotitic melt is somewhere in the range between 0.01 and 0.1 Pa*s, perhaps ~0.05 Pa*s (50 mPa*s). This value falls in the range of H₂O-irch (and Fe-free) melt viscosities reported in the manuscript under review. This, I find strange because H₂O-bearing melts should have much lower viscosities at everything else being equal. The inconsistency perhaps can be explained by that I am comparing the viscosities of an Fe-bearing and Fe-free melts, but the authors argue that "Since Fe should have little effect on the viscosity of 410 melt, an Fe-free starting material was prepared to avoid the density change due to Fe loss to the Pt capsule lid." at lines 42-44 of their Supplementary Material. I find to accept this statement without references given that there is robust experimental data that show a strong effect of Fe and its oxidation state on the melt viscosity at 1 atm, e.g.: <https://www.sciencedirect.com/science/article/pii/0016703787902316>

If one accepts that Fe and its oxidation state is an important factor that can control melt viscosity, then the relevance and value of the presented viscosities of Fe-free melts is unclear to me because all silicates melts in the mantle contain iron.

Similarly, a comparison to Fig.7 of Russell et al. (2024) shows that the results of the reviewed paper are, at least partially, inconsistent with this new published model for ultramafic melts. According to Russell et al., (2024), a dry ultramafic melt at ~14 GPa at liquidus T should have the viscosity of 60-70 mPa*s. This is consistent with Xie et al., (2021), but clearly inconsistent with the results of this work that suggest that a melt with ~15mol% H₂O has the viscosity of ~96 mPa*s. I recall here again that a melt with higher H₂O content should have a lower viscosity and not the other way around.

I recommend that the authors present a thorough comparison with the extant viscosity models and (at the very least) offer explanations for the above issues and other discrepancies they can identify.

Minor: According to the main text (e.g. Lines 98-99), melts with three distinct H₂O-concentrations were studied. Yet, the supplementary material lists four distinct H₂O concentrations (e.g. Table 1 in the supplement)?

Reviewer #2

(Remarks to the Author)

This manuscript reports the geophysical models for the formations of the single melt layers and the melt doublets in the lower part of the upper mantle based on their results of the high-pressure experiments and the 1D finite element simulations. The authors determined viscosity of hydrous silicate melts at high pressure relevant to the bottom of the upper mantle by performing the falling sphere experiments using the multi-anvil apparatus with synchrotron X-ray measurements. Then they simulated the melt distribution in the lower part of the upper mantle using viscosity of the hydrous silicate melts. The results and discussion of this study are interesting and important. I recommend accepting this manuscript for publication after considering the following suggestions.

[1] Units of water content (Figs. 1 and 2; L. 34, 99, and so on)

It is better to use "wt%" as the unit of water content for readers instead of "mol%".

Other suggestions:

L. 4: zhao -> Zhao

L. 78: 0. 2 -> 0.2

L. 105: Table -> Tables

L. 108, Figs. 1 and 4: Pa.s -> Pa·s

Fig. 1: 2020 -> (2020)

Fig. 1: 2021 -> (2021)

L. 132: first principle -> first principle (FP)

L. 209: model -> models

L. 368: patterns -> measurements

L. 369: SPring 8 -> SPring-8

L. 371: ?? -> numbers

L. 373: XX -> numbers

L. 376: ?? -> fund name

<Supplementary materials>

L. 4: zhao -> Zhao

L. 29: 2010 -> (2010)

L. 35: 2015 -> (2015)

L. 49: wt.% -> wt%

L. 59: 2020 -> (2020)

L. 77: 11W -> 11 W

L. 77: 40kHz -> 40 kHz

L. 96: 0.1s -> 0.1 s

L. 131: 300K -> 300 K

Supplementary Table 1, L. 251: Pa.s -> Pa·s

L. 147: ps, -> ps, pm

L. 150: EOS -> equation of state

Supplementary Table 3: wt.% -> wt%

L. 211: 2020 -> (2020)

L. 224: Kelvin -> K

L. 248: . -> :

L. 282: Eq. -> Eqs.

L. 287: Where -> , where

L. 328: decay -> decay.

L. 340: Fig.14 -> Fig. 14

L. 363: Fig.15 -> Fig. 15

L. 364: (IV -> IV

Reviewer #3

(Remarks to the Author)

Xie et al. reports the measured data for viscosity of hydrous silicate melt around 14 GPa in the temperature range 1873 to 2673 K. Their finding of relatively low melt viscosity values around 14 GPa are consistent with the previously measured and calculated results for both dry and hydrous silicate melts. Moreover, they have used their new viscosity data along with several other physical quantities to reassess the validity of the advective

thickening model and to explore melt distribution by performing 1D finite element simulations.

Their predicted melt distributions, including melt doublet appear to be consistent with the seismic observations of low velocities.

The authors might want to address some important concerns as follows.

Several previous studies have suggested possible existence of hydrous melt layer(s) at 410 km depth. The authors need to talk about those studies, such as Karato et al., PEPS 7:76, 2020; Mookherjee et al., Nature 452: 983, 2008.

Fig. 1 shows the measured viscosity data of 410 melt.

Referring the melt composition of their experimental setup as "410 melt" may not be a good idea. For the sake of clarity, it is better to call the melt studied as hydrous silicate melt or hydrous peridotitic melt. It is helpful to give the actual melt compositions corresponding to three data points (shown in Fig 1) in the paper although the relevant information can be found in supplementary file.

Only three data values for melt viscosity were measured using in-situ falling sphere viscometry. As such, it is not easy to interpret such a few data and justify their validity, for instance, isolating the effects of temperature versus water is confusing. The inferred viscosity value of dry melt is higher by one order of magnitude than the existing measured data for both dry forsterite and peridotite melts (Fig 1).

The authors should consider more data from the literature for comparison. How about adding in Fig 1 the viscosity values of dry melt corresponding to each of three temperatures 1700, 1800 and 2000 degrees (14 GPa) of hydrous melt measurements. These values can be obtained from Ref. 20 (Xie et al., 2021), Ref. 21 (Xie et al. 2022), and Ref. 26 (Cochain et al., 2017) of the paper. This way one can see how much the viscosity drops for three different amounts of water as reported here.

First-principles results from Ref. 23 (Drewitt et al., 2022) are used for comparison. There exist more first-principles studies of silicate melt viscosity (Ref. 22: Huang et al., 2024; Bajgain et al., 2022), including hydrous melt (Karki and Stixrude, 2010), which all have predicted relatively low viscosities for silicate melts, agreeing with this study.

For instance, the calculated viscosity of hydrous enstatite melt (containing 10 wt.% or 25 mol% water) is around 0.015 Pa.s at 14 GPa and 2200 K (see Fig 3 of the paper by Karki and Stixrude, Science, 2010). This value compares favorably with the second data point in Fig. 1. Similarly, showing the calculated value of about 0.02 Pa.s for dry peridotitic melt (Huang et al., 2024) in Fig 1 is important.

The authors argue that the AT model fails to explain the steep lateral variation of layer thickness and the formation of melt doublets.

Their 1D finite element simulations have predicted melt layering, including doublets that can be linked to seismic low velocity layers. The possible two distinct melt layers appear to arise mainly because of continuous dehydration melting (CDM) consideration which perhaps causes the neutral density position (NDP) at depths shallower than the 410 km. It has less to do with low melt viscosities. Is this true?

"A melt-mantle density contrast of 0.08-1.3 kg/m³ at a depth of 410 km results in a neutral melt-mantle density position at an elevation of only ~0.08-1.3 km above the 410." How do you infer the elevation range?

The authors presented and tested their proposed model in details (Fig 2-4) by exploring the model sensitiveness with respect to all physical parameters. Their findings have important implications for melt distributions at or above 410 km.

Version 1:

Reviewer comments:

Reviewer #1

(Remarks to the Author)

I find that the revised manuscript and the rebuttal letter adequately addressed the issues raised in my review. I thank the authors for their effort to improve the manuscript, and I am happy to recommend this manuscript for publication in Nature Communications.

Reviewer #3

(Remarks to the Author)

The authors have revised the paper in a significant way by taking into account all reviewers' comments and suggestions. Their responses mostly make sense. While a relatively few measured data points are considered, the comparisons with other relevant data in the literature look encouraging. The inferred implication for the seismic observations of possible melt layers is quite important. I believe that the paper is worth publishing.

Revision information

First of all, we sincerely appreciate the editor and reviewers for their evaluation and comments on our manuscript [NCOMMS-24-80744] entitled “Low Melt Viscosity Enables Melt Doublets Above The 410-km Discontinuity”.

The following is a list of our revisions or responses:

Reviewer #1 (Remarks to the Author): (our responses are in brown and quotes from the main text are in red)

I. I have now reviewed the manuscript “Low Melt Viscosity Enables Melt Doublets Above The 410-km Discontinuity” by Xie et al., submitted to Nature Communications. This study address the origin of the seismic discontinuities at 300-410 km depths in the Earth’s mantle, for which partial melts may provide a viable explanation. In this work, the authors performed extremely challenging high-pressure experiments to measure the viscosity of hydrous-bearing iron-free silicate melts, and then used the results together with other available physical information on the properties of mantle melts to model melt segregation above the 410 km depth.

Overall, this is a very thorough study and I applaud the authors for their effort. Yet, my background is in physical properties of glasses and melts, and I am unable to judge the technical quality of the numerical simulations performed here (to me they seem sufficiently comprehensive). I hope another reviewer will provide their expert opinion on the quality of numerical simulations. The experimental part, however, has issues and can be improved. After these issues are resolved, the paper may become suitable for publication in Nature Communications.

Thank you for your thorough review and for recognizing the effort involved in our study. We greatly appreciate your positive feedback regarding the experimental work and your acknowledgment of the comprehensiveness of the numerical simulations. Below, we address the specific issues and suggestions raised regarding the experimental portion of the manuscript to improve its clarity and robustness.

II. My main concern lies with the reliability of the presented viscosity measurements, which is difficult to assess by comparing to reference materials with constrained viscosity at similar conditions. The ability of the used experimental technique to reproduce literature data is not demonstrated in the present manuscript. A corresponding (i.e., Fe-free) dry ultramafic melt was not probed here, although this could have allowed an independent check on the reliability of the viscosity measurements.

Thank you for your thoughtful comments and for raising concerns regarding the reliability of the viscosity measurements. Below, we address your points in detail to clarify and demonstrate the robustness of our experimental approach:

1. The ability of our experimental technique to reproduce literature data has been demonstrated in our previous work:

The technique employed in this study is identical to that in our previous publications (Xie et al., Nature Communications, 2020; Xie et al., GRL, 2021). The reproducibility of the method is within 6%, which is documented in Xie et al. 2020. The only differences in this study are the

capsule material (graphite in previous work and diamond in this work) and sample composition, both of which have no influence on the reliability of the technique.

2. The difficulty to measure viscosity of dry 410 melt using current capsule design:

While we acknowledge that measurements on a corresponding Fe-free dry ultramafic melt could provide additional validation, conducting such an experiment in the same capsule design is impractical due to the limitations of the design. Specifically, the melting point of Pt (~2100°C), used to seal the melt in our capsule design, is lower than the melting temperature of a dry ultramafic sample (over 2400°C). As such, a dry ultramafic melt experiment would face inherent technical challenges, such as capsule failure, rendering it infeasible with the current experimental design.

3. Validation of our results using literature data:

To address concerns regarding reliability, we have updated Fig. 1 to include the extrapolated viscosity of dry silicates at the same pressure and temperature conditions as our experiments. As shown in the revised figure, our viscosity data align well with previously measured and calculated results for both dry and hydrous silicate melts. Discrepancies highlighted by Reviewer 1 are attributed to specific effects, such as temperature or Ca content, and are explained in detail in subsequent responses.

III. Below, I examine the presented results by comparing them to authors' own data on the viscosity of an Fe-bearing dry peridotitic melt (Xie et al., GRL, 2021) but also to the new model of Russell et al., 2024 (not cited in this work, but certainly relevant): <https://www.sciencedirect.com/science/article/pii/S0012821X24003327>

One problem I see is that the previous study from the same group and using the same method (Xie et al., GRL, 2021) suggests that the viscosity of an Fe-bearing and dry peridotitic melt is somewhere in the range between 0.01 and 0.1 Pa*s, perhaps ~0.05 Pa*s (50 mPa*s). This value falls in the range of H₂O-irch (and Fe-free) melt viscosities reported in the manuscript under review. This, I find strange because H₂O-bearing melts should have much lower viscosities at everything else being equal.

Following the suggestion, we updated Fig. 1 by incorporating literature data. Below, we provide a detailed explanation of the updates made to Fig. 1 and address the points you raised:

1. Updated Figure with Literature Data:

We have updated Fig. 1 to include literature data, specifically the extrapolated viscosity of dry silicates based on models from Xie et al. (2020, 2021) and Russell et al. (2024), calculated under the same pressure-temperature conditions as our experiments. This additional context provides a clearer basis for comparing our results with existing data.

“

Fig. 1. Viscosity of hydrous peridotitic melt along its melting curve as a function of water content. Red circles represent experimental data from this study with the red solid line representing a linear fit of the logarithmic viscosity as a function of water content. The red dashed line indicates the linear extrapolation to dry conditions. Gray circles indicate predictions based on the Vogel-Fulcher-Tammann (VFT) model of hydrous peridotite from Russel et al. (2024)³⁹. To isolate the effects of compositions and temperature, extrapolated viscosity of dry super-cooled silicate liquids—including diopside²³, enstatite²³, forsterite²³, peridotite²² and 410 melt compositions³⁴—are shown as colored dashed lines, calculated under the same pressure-temperature conditions as the experiments in this study. Since the water content in the dry super-cooled silicate liquids is zero, the lower horizontal axis is invalid for them. The viscosity of dry 410 melt was derived using an Arrhenius equation for forsterite melt with Ca-corrected activation enthalpy (see Suppl. Materials for details). The gray dashed line with an error envelope represents viscosity estimates for hydrous peridotitic melt based on first principle (FP) calculation²⁶. The gray square marks the viscosity of liquid hydrous enstatite²⁸, while the gray diamond indicates the viscosity of dry peridotite derived from FP calculations²⁴. D_Di: dry diopside, D_En: dry enstatite, D_Fo: dry forsterite, D_PE: dry peridotite, D_410 melt: dry 410 melt composition from Xie et al. submitted³⁴, H_En: hydrous enstatite, H_PE: hydrous peridotite.”

2. Liquidus Considerations:

Our experiments were conducted along the liquidus of the 410 melt. The presence of H₂O lowers the liquidus temperature and alters the melt structure. These changes have competing effects on viscosity:

- Temperature Effect: Lower temperatures tend to increase viscosity.
- Structural Effect: Changes in melt structure due to H₂O typically reduce viscosity, as noted by Reviewer 1.

Thus, the observed viscosity values reflect the interplay between these two effects. It is not unexpected that the viscosity of dry peridotitic melt along a dry liquidus overlaps with the viscosity of hydrous 410 melt along a wet liquidus.

3. Comparisons under the same P-T conditions:

Direct comparisons between viscosities measured along wet and dry liquidus lines are indeed inappropriate. Instead, we have compared the extrapolated viscosity of supercooled dry peridotite melt along a wet liquidus, which is calculated according to the parameterized Arrhenius equations from Xie et al. (2020 and 2021), with our experimental data for hydrous 410 melt.

As shown in the updated Fig. 1, the viscosity of hydrous 410 melt along its wet liquidus is generally lower than the extrapolated viscosity of supercooled dry peridotite melt under identical P-T conditions, as the reviewer expects, with the exception of the point at ~15 mol% H₂O.

4. Role of Ca Content to explain the point at ~15 mol% H₂O:

As shown in Suppl. Table 1, the sample in this study is significantly Ca-rich compared to typical peridotite compositions. The high Ca content of the 410 melt has a significant impact on its viscosity. For instance, diopside melt, which is Ca-rich, exhibits much higher viscosity than enstatite melt. In the updated Fig. 1, we include the extrapolated viscosity of 410 melt with a Ca-effect correction (see details of the extrapolation method in Section 2 of Suppl. Materials), which is significantly higher than all the measured viscosity of hydrous 410 melt under the same conditions, showing the reliability of the point at ~15 mol% H₂O.

Therefore, the apparent "discrepancy" noted by Reviewer 1 can be attributed primarily to the combined effects of temperature and Ca content. Our analysis demonstrates that the results are consistent with theoretical expectations once these effects are considered.

Related information was added in Lines 113-116:

“The elemental composition of the 410 melt was determined in our recent work³⁴ and was used as the basis for the starting materials, excluding H₂O. Its dry composition includes SiO₂, Al₂O₃, (Mg,Fe)O, and CaO at molar ratios of 29.9, 0.35, 53.6, and 16.2, respectively, which has a much higher Ca content than typical peridotite.”

And in Lines 140-149:

“Compared to the extrapolated viscosity of supercooled dry peridotite melt at the same experimental pressure-temperature conditions, the measured viscosities of the hydrous samples are generally lower, with the exception of the point at approximately 15 mol% H₂O. As illustrated by the higher viscosity of diopside melt compared to enstatite melt (Fig. 1), the presence of Ca can significantly increase melt viscosity. Since the sample composition in this study is much richer in Ca than typical peridotite compositions, the higher viscosity observed at ~15 mol% H₂O can be

attributed to the Ca effect. After correcting for the Ca effect, the extrapolated viscosity of dry samples is significantly higher than all the measured viscosities of hydrous samples under the same conditions, which further validates the reliability of our measurements.”

IV. The inconsistency perhaps can be explained by that I am comparing the viscosities of an Fe-bearing and Fe-free melts, but the authors argue that “Since Fe should have little effect on the viscosity of 410 melt, an Fe-free starting material was prepared to avoid the density change due to Fe loss to the Pt capsule lid.” at lines 42-44 of their Supplementary Material. I find to accept this statement without references given that there is robust experimental data that show a strong effect of Fe and its oxidation state on the melt viscosity at 1 atm,

e.g.: <https://www.sciencedirect.com/science/article/pii/S0016703787902316>

If one accepts that Fe and its oxidation state is an important factor that can control melt viscosity, then the relevance and value of the presented viscosities of Fe-free melts is unclear to me because all silicates melts in the mantle contain iron.

1. Fe content has little effect on the viscosity of silicate melt at high pressure:

As shown in the Supplementary Fig. 6 in Xie et al. (2020), Fe does have a significant effect on the viscosity of melt at low pressures. However, the Fe effect decreases with increasing pressure and almost diminishes above 10 GPa by comparing the viscosity of liquid Fo (Xie et al. 2020) and Fa (Spice et al. 2015).

[Figure Redacted]

2. The valence state of Fe has little effect on the viscosity of 410 melt:

Experimental studies (e.g., Dingwell and Virgo, 1987) indicate that the viscosity of silicate melts decreases with the reduction of Fe, but the effect becomes negligible at moderate to low Fe³⁺/total Fe ratios (<0.4), where a region of viscosity invariance is observed.

The activity of H₂O can influence oxygen fugacity (fO₂) through the dissociation reaction (H₂O ↔ H₂ + 1/2 O₂), which in turn affects the valence state of Fe in Fe-bearing silicate melts (Botcharnikov et al., GCA, 2005, DOI: <https://doi.org/10.1016/j.gca.2005.04.023>). In the upper mantle, however, the fO₂ is buffered by the quartz-fayalite-magnetite (QFM) system because the quantity of hydrous silicate melt generated under these conditions is tiny. According to Fig.11 in

Botcharnikov et al. (2005), the Fe^{3+} /total Fe ratio in these melts is approximately 0.15, which is well below the threshold of 0.4. Consequently, the valence state of Fe in hydrous silicate melts generated in the upper mantle likely has little to no impact on their viscosity.

[Figure Redacted]

Fig. 11 from Botcharnikov et al., GCA (2005)

In summary, both the Fe content and its valence state have little effect on the viscosity of 410 melt. The viscosity of Fe-free melt is a good approximation of the Fe-bearing 410 melt.

We modified the following sentence in main text (lines 151-154) as:

“At the conditions of pressure and temperature relevant to this study, neither Fe content^{23,36} nor its valence state^{37,38} have a significant effect on the viscosity of 410 melts (see detailed discussion in section 1.1 of Suppl. Materials).”

We added the following references:

36. Spice, H., Sanloup, C., Cochain, B., de Grouchy, C. & Kono, Y. Viscosity of liquid fayalite up to 9 GPa. *Geochim Cosmochim Acta* **148**, 219–227 (2015).
37. Botcharnikov, R. E., Koepke, J., Holtz, F., McCammon, C. & Wilke, M. The effect of water activity on the oxidation and structural state of Fe in a ferro-basaltic melt. *Geochim Cosmochim Acta* **69**, 5071–5085 (2005).
38. Dingwell, D. B. & Virgo, D. The effect of oxidation state on the viscosity of melts in the system Na₂O-FeO-Fe₂O₃-SiO₂. *Geochim Cosmochim Acta* **51**, 195–205 (1987).

We modified the sentence in the Supplementary Materials (lines 43-58) as follows:

“ Although Fe content has a significant effect on the viscosity of silicate melt at low pressures, it should have little effect on the viscosity of 410 melt at pressures above 10 GPa^{5,6}. An experimental study⁷ demonstrated that the valence of Fe has little effect on the viscosity of silicate melts at moderate to low values of Fe³⁺/total Fe (<0.4). The activity of H₂O can influence oxygen fugacity (fO₂) through the dissociation reaction (H₂O ↔ H₂ + 1/2 O₂), which in turn affects the valence state of Fe in Fe-bearing silicate melts⁸. In the upper mantle, however, the fO₂ is controlled by the quartz-fayalite-magnetite (QFM) system because the quantity of hydrous silicate melt generated under these conditions is tiny. The upper mantle f(O₂) is at or lower than the QFM buffer and gets more reducing with depth⁹. Estimates suggest that even at subduction zones, f(O₂) is 1-2 log units below QFM by 200 km depth⁹. The Fe³⁺/total Fe ratio in silicate melts is approximately 0.15 in a QFM buffered upper mantle⁸, which is well below the threshold of 0.4. Therefore, the viscosity of Fe-free 410 melt is a good approximation of that of 410 melt. In light of this, and in order to avoid the density change due to Fe loss to the Pt capsule lid, Fe-free starting materials were used.”

We also added the following references:

6. Spice, H., Sanloup, C., Cochain, B., de Grouchy, C. & Kono, Y. Viscosity of liquid fayalite up to 9 GPa. *Geochim Cosmochim Acta* **148**, 219–227 (2015).
7. Dingwell, D. B. & Virgo, D. The effect of oxidation state on the viscosity of melts in the system Na₂O-FeO-Fe₂O₃-SiO₂. *Geochim Cosmochim Acta* **51**, 195–205 (1987).
8. Botcharnikov, R. E., Koepke, J., Holtz, F., McCammon, C. & Wilke, M. The effect of water activity on the oxidation and structural state of Fe in a ferro-basaltic melt. *Geochim Cosmochim Acta* **69**, 5071–5085 (2005).

V. Similarly, a comparison to Fig.7 of Russell et al. (2024) shows that the results of the reviewed paper are, at least partially, inconsistent with this new published model for ultramafic melts. According to Russell et al., (2024), a dry ultramafic melt at ~14 GPa at liquidus T should have the viscosity of 60-70 mPa*s. This is consistent with Xie et al., (2021), but clearly inconsistent with the results of this work that suggest that a melt with ~15mol% H₂O has the viscosity of ~96 mPa*s. I recall here again that a melt with higher H₂O content should have a lower viscosity and not the other way around.

We note here that the Ca content of our sample is much higher than a pyrolite composition (see **response III**). As shown in the updated Fig. 1, the viscosity of diopside melt is much higher than that of enstatite melt at 14 GPa, indicating a strong effect of Ca on the viscosity. This apparent “discrepancy” should be due to the Ca-effect.

VI. I recommend that the authors present a thorough comparison with the extant viscosity models and (at the very least) offer explanations for the above issues and other discrepancies they can identify.

Following this suggestion, we present a thorough comparison with the extant viscosity models in the updated Fig.1 by adding the extrapolated viscosity of super-cooled dry liquid compositions and other hydrous viscosities from the literature.

Detailed explanations can be found in **response III**.

VII. Minor: According to the main text (e.g. Lines 98-99), melts with three distinct H₂O-concentrations were studied. Yet, the supplementary material lists four distinct H₂O concentrations (e.g. Table 1 in the supplement)?

In our experiments, we tested four distinct H₂O concentrations. However, for the experiment with the highest water content (39.7 mol%), the falling sphere failed to reach terminal velocity despite repeated attempts across more than three experiments. Initially, we considered that these data might provide useful insights for readers, such as an upper bound on viscosity. However, upon reflection, it appears that including these results causes more confusion than clarity. In this version of the manuscript, we have removed this data.

Accordingly, we removed the related information in Suppl. Table 1, Suppl. Fig. 5, Suppl. Table 2, and Suppl. Table 3.

Reviewer #2 (Remarks to the Author):

VIII. This manuscript reports the geophysical models for the formations of the single melt layers and the melt doublets in the lower part of the upper mantle based on their results of the high-pressure experiments and the 1D finite element simulations. The authors determined viscosity of hydrous silicate melts at high pressure relevant to the bottom of the upper mantle by performing the falling sphere experiments using the multi-anvil apparatus with synchrotron X-ray measurements. Then they simulated the melt distribution in the lower part of the upper mantle using viscosity of the hydrous silicate melts. The results and discussion of this study are interesting and important. I recommend accepting this manuscript for publication after considering the following suggestions.

We thank Reviewer 2 for the positive feedback on our manuscript and the recognition of the significance and relevance of our study. We appreciate the encouraging recommendation for publication and carefully addressed the suggestions provided to further improve the quality and clarity of the manuscript.

IX. [1] Units of water content (Figs. 1 and 2; L. 34, 99, and so on)

It is better to use “wt%” as the unit of water content for readers instead of “mol%”.

We deliberated extensively on whether to use wt% or mol% as the unit for H₂O content in the manuscript. The Fe content in hydrous melts can vary significantly within the mantle, leading to notable changes in the wt% of H₂O for the same mol%. However, the viscosity of the melt remains relatively unaffected by such changes in Fe content. Ultimately, we chose mol% as it more accurately captures the intrinsic effect of water on melt properties, providing a clearer and more consistent basis for interpretation.

X. Other suggestions:

L. 4: zhao -> Zhao

followed and corrected

L. 78: 0. 2 -> 0.2

followed and corrected

L. 105: Table -> Tables

followed and corrected

L. 108, Figs. 1 and 4: Pa.s -> Pa·s

followed and corrected

Fig. 1: 2020 -> (2020)

followed and corrected

Fig. 1: 2021 -> (2021)

followed and corrected

L. 132: first principle -> first principle (FP)

followed and corrected

L. 209: model -> models

followed and corrected

L. 368: patterns -> measurements

followed and corrected

L. 369: SPring 8 -> SPring-8

followed and corrected

L. 371: ?? -> numbers:

followed and corrected

L. 373: XX -> numbers

XX itself is the number.

L. 376: ?? -> fund name

followed and corrected

<Supplementary materials>

L. 4: zhao -> Zhao

followed and corrected

L. 29: 2010 -> (2010)

followed and corrected

L. 35: 2015 -> (2015)

followed and corrected

L. 49: wt.% -> wt%

followed and corrected

L. 59: 2020 -> (2020)

followed and corrected
L. 77: 11W -> 11 W
followed and corrected
L. 77: 40kHz -> 40 kHz
followed and corrected
L. 96: 0.1s -> 0.1 s
followed and corrected
L. 131: 300K -> 300 K
followed and corrected
Supplementary Table 1, L. 251: Pa.s -> Pa·s
followed and corrected
L. 147: ρ s, -> ρ s, ρ m
followed and corrected
L. 150: EOS -> equation of state
followed and corrected
Supplementary Table 3: wt.% -> wt%
followed and corrected
L. 211: 2020 -> (2020)
followed and corrected
L. 224: Kelvin -> K
followed and corrected
L. 248: . -> :
followed and corrected
L. 282: Eq. -> Eqs.
followed and corrected
L. 287: Where -> , where
followed and corrected
L. 328: decay -> decay
followed and corrected.
L. 340: Fig.14 -> Fig. 14
followed and corrected
L. 363: Fig.15 -> Fig. 15
followed and corrected
L. 364: (IV -> IV
followed and corrected

Reviewer #3 (Remarks to the Author):

XI. Xie et al. reports the measured data for viscosity of hydrous silicate melt around 14 GPa in the temperature range 1873 to 2673 K. Their finding of relatively low melt viscosity values around 14 GPa are consistent with the previously measured and calculated results for both dry and hydrous silicate melts. Moreover, they have used their new viscosity data along with several other physical quantities to reassess the validity of the advective

thickening model and to explore melt distribution by performing 1D finite element simulations.

Their predicted melt distributions, including melt doublet appear to be consistent with the seismic observations of low velocities.

The authors might want to address some important concerns as follows.

We thank Reviewer 3 for their positive comments on our work and for recognizing the significance of our findings. We appreciate the acknowledgment of our viscosity measurements, their consistency with previous studies, and the relevance of our 1D finite element simulations in providing insights into melt distribution and its alignment with seismic observations. We have carefully addressed the important concerns raised in the following responses.

XII. Several previous studies have suggested possible existence of hydrous melt layer(s) at 410 km depth. The authors need to talk about those studies, such as Karato et al., PEPS 7:76, 2020; Mookherjee et al., Nature 452: 983, 2008.

Following this suggestion, we have added the following citations in lines 68-69: "These LVLs are frequently explained by the presence of up to a few wt% of hydrous silicate melt (hereafter referred to as "410 melt")^{1-3,11,12}."

XIII. Fig. 1 shows the measured viscosity data of 410 melt. Referring the melt composition of their experimental setup as "410 melt" may not be a good idea. For the sake of clarity, it is better to call the melt studied as hydrous silicate melt or hydrous peridotitic melt.

Following this suggestion, the melt composition in our experiments is referred to as "hydrous peridotitic melt" at lines 117,135, 176.

XIV. It is helpful to give the actual melt compositions corresponding to three data points (shown in Fig 1) in the paper although the relevant information can be found in supplementary file.

Following this suggestion, we added the composition of our sample in the main text (lines 113-116) as follows:

"The elemental composition of the 410 melt was determined in our recent work³⁴ and was used as the basis for the starting materials, excluding H₂O. Its dry composition includes SiO₂, Al₂O₃, (Mg,Fe)O, and CaO at molar ratios of 29.9, 0.35, 53.6, and 16.2, respectively, which has a much higher Ca content than typical peridotite."

XV. Only three data values for melt viscosity were measured using in-situ falling sphere viscometry. As such, it is not easy to interpret such a few data and justify their validity, for instance, isolating the effects of temperature versus water is confusing. The inferred viscosity value of dry melt is higher by one order of magnitude than the existing measured

data for both dry forsterite and peridotite melts (Fig 1).

The authors should consider more data from the literature for comparison. How about adding in Fig 1 the viscosity values of dry melt corresponding to each of three temperatures 1700, 1800 and 2000 degrees (14 GPa) of hydrous melt measurements. These values can be obtained from Ref. 20 (Xie et al., 2021), Ref. 21 (Xie et al. 2022), and Ref. 26 (Cochain et al., 2017) of the paper. This way one can see how much the viscosity drops for three different amounts of water as reported here.

First-principles results from Ref. 23 (Drewitt et al., 2022) are used for comparison. There exist more first-principles studies of silicate melt viscosity (Ref. 22: Huang et al., 2024; Bajgain et al., 2022), including hydrous melt (Karki and Stixrude, 2010), which all have predicted relatively low viscosities for silicate melts, agreeing with this study.

For instance, the calculated viscosity of hydrous enstatite melt (containing 10 wt.% or 25 mol% water) is around 0.015 Pa.s at 14 GPa and 2200 K (see Fig 3 of the paper by Karki and Stixrude, Science, 2010). This value compares favorably with the second data point in Fig. 1. Similarly, showing the calculated value of about 0.02 Pa.s for dry peridotitic melt (Huang et al., 2024) in Fig 1 is important.

Following this suggestion, we have updated Fig. 1 (lines 175-200) to include literature data, adding the extrapolated viscosity of dry silicates based on models from Xie et al. (2020, 2021), Russell et al. (2024), Karki and Stixrude (2010), and Huang et al. (2024). Related comparisons are added in lines 140-149:

“Compared to the extrapolated viscosity of supercooled dry peridotite melt at the same experimental pressure-temperature conditions, the measured viscosities of the hydrous samples are generally lower, with the exception of the point at approximately 15 mol% H₂O. As illustrated by the higher viscosity of diopside melt compared to enstatite melt (Fig. 1), the presence of Ca can significantly increase melt viscosity. Since the sample composition in this study is much richer in Ca than typical peridotite compositions, the higher viscosity observed at ~15 mol% H₂O can be attributed to the Ca effect. After correcting for the Ca effect, the extrapolated viscosity of dry samples is significantly higher than all the measured viscosities of hydrous samples under the same conditions, which further validates the reliability of our measurements.”

Fig. 1. Viscosity of hydrous peridotitic melt along its melting curve as a function of water content. Red circles represent experimental data from this study with the red solid line representing a linear fit of the logarithmic viscosity as a function of water content. The red dashed line indicates the linear extrapolation to dry conditions. Gray circles indicate predictions based on the Vogel-Fulcher-Tammann (VFT) model of hydrous peridotite from Russel et al. (2024)³⁹. To isolate the effects of compositions and temperature, extrapolated viscosity of dry super-cooled silicate liquids—including diopside²³, enstatite²³, forsterite²³, peridotite²² and 410 melt compositions³⁴—are shown as colored dashed lines, calculated under the same pressure-temperature conditions as the experiments in this study. Since the water content in the dry super-cooled silicate liquids is zero, the lower horizontal axis is invalid for them. The viscosity of dry 410 melt was derived using an Arrhenius equation for forsterite melt with Ca-corrected activation enthalpy (see Suppl. Materials for details). The gray dashed line with an error envelope represents viscosity estimates for hydrous peridotitic melt based on first principle (FP) calculation²⁶. The gray square marks the viscosity of liquid hydrous enstatite²⁸, while the gray diamond indicates the viscosity of dry peridotite derived from FP calculations²⁴. D_Di: dry diopside, D_En: dry enstatite, D_Fo: dry forsterite, D_PE: dry peridotite, D_410 melt: dry 410 melt composition from Xie et al. submitted³⁴, H_En: hydrous enstatite, H_PE: hydrous peridotite.”

The reference to Bajgain et al., 2022 is added at line 102:

“The melt viscosity used was 1 Pa·s, whereas the viscosity of dry basaltic/peridotitic melts under high pressures could be as low as ~10 mPa·s^{22–25}, with water further reducing their viscosity^{26–28}”

XVI. The authors argue that the AT model fails to explain the steep lateral variation of layer thickness and the formation of melt doublets.

Their 1D finite element simulations have predicted melt layering, including doublets that can be linked to seismic low velocity layers. The possible two distinct melt layers appear to arise mainly because of continuous dehydration melting (CDM) consideration which perhaps causes the neutral density position (NDP) at depths shallower than the 410 km. It has less to do with low melt viscosities. Is this true?

We appreciate the reviewer's detailed comments and the opportunity to clarify the role of low melt viscosity in our model.

To address your question, it is not accurate to suggest that low melt viscosity has little to do with the formation of the melt doublets. The melt doublet model involves three critical components: (1) the top melt layer at the neutral density position (NDP), (2) a nearly melt-free gap, and (3) the bottom melt layer at the 410-km discontinuity. Continuous dehydration melting (CDM) is indeed essential for forming the top melt layer at the NDP by supplying a continuous source of melt. Similarly, the density trap at 410 km plays a pivotal role in forming the bottom melt layer. However, the key factor enabling the formation of the melt-free gap between the layers is the low viscosity of the melt.

For instance, if a higher viscosity value (e.g., 1 Pa·s) is used, as assumed in previous studies, the advective thickening (AT) model remains valid, and only a single, thick melt layer forms. In contrast, the extremely low viscosity values we report ensure rapid melt segregation, preventing advection from forming a single thick layer and enabling the formation of two distinct melt layers. Thus, low viscosity is a crucial precondition for the emergence of melt doublets in our model.

XVII. "A melt-mantle density contrast of 0.08-1.3 kg/m³ at a depth of 410 km results in a neutral melt-mantle density position at an elevation of only ~0.08–1.3 km above the 410." How do you infer the elevation range?

In the advective thickening (AT) model, the melt layer extends from the neutral density position (NDP) to the maximum depth where the mantle upwelling is able to drag the melt upward. At a given viscosity, this maximum depth depends on the density contrast between the melt and the surrounding mantle, as governed by Darcy's law. Based on our measured viscosity values, the mantle upwelling can drag melt upward when the density contrast is within the range of 0.08–1.3 kg/m³. At the NDP itself, the density contrast is effectively zero.

The maximum thickness of the melt layer is thus determined by the depth range over which the density contrast varies from 0 to 0.08–1.3 kg/m³. As shown in Supplementary Fig. 10, the relationship between density contrast and depth is nearly linear over a short depth interval. Using this linear slope, we calculate the corresponding depth interval for the given density contrast range, which yields the elevation range of ~0.08–1.3 km above the 410-km discontinuity. This calculation captures the maximum possible elevation range for the melt layer in the context of the AT model.

XVIII. The authors presented and tested their proposed model in details (Fig 2-4) by exploring the model sensitiveness with respect to all physical parameters. Their findings have important implications for melt distributions at or above 410 km.

We thank Reviewer 3 for the positive comments and recognition of the detailed model exploration presented in Figures 2–4. We are pleased that the reviewer finds our findings to have important implications for melt distributions at or above 410 km.